# A Computational Separation between Private Learning and Online Learning*

**Mark Bun**
Department of Computer Science
Boston University
Boston, MA 02215
mbun@bu.edu

## Abstract

A recent line of work has shown a qualitative equivalence between differentially private PAC learning and online learning: A concept class is privately learnable if and only if it is online learnable with a finite mistake bound. However, both directions of this equivalence incur significant losses in both sample and computational efficiency. Studying a special case of this connection, Gonen, Hazan, and Moran (NeurIPS 2019) showed that uniform or highly sample-efficient pure-private learners can be time-efficiently compiled into online learners. We show that, assuming the existence of one-way functions, such an efficient conversion is impossible even for general pure-private learners with polynomial sample complexity. This resolves a question of Neel, Roth, and Wu (FOCS 2019).

## 1 Introduction

Differential privacy [Dwork et al., 2006] offers formal guarantees and a rich algorithmic toolkit for studying how such analyses can be conducted. To address settings where sensitive data arise in machine learning, Kasiviswanathan et al. [2011] introduced differentially private PAC learning as a privacy-preserving version of Valiant's PAC model for binary classification Valiant [1984]. In the ensuing decade a number of works (e.g., Beimel et al. [2014], Bun et al. [2015], Feldman and Xiao [2015], Beimel et al. [2016, 2019], Alon et al. [2019], Kaplan et al. [2020]) have developed sophisticated algorithms for learning fundamental concept classes while exposing deep connections to optimization, online learning, and communication complexity along the way.

Despite all of this attention and progress, we are still far from resolving many basic questions about the private PAC model. As an illustration of the state of affairs, the earliest results in *non-private* PAC learning showed that the sample complexity of (i.e., the minimum number of samples sufficient for) learning a concept class $\mathcal{C}$ is tightly characterized by its VC dimension [Vapnik and Chervonenkis, 1974, Blumer et al., 1989]. It is wide open to obtain an analogous characterization for private PAC learning. In fact, it was only in the last year that a line of work [Bun et al., 2015, Alon et al., 2019, Bun et al., 2020b] culminated in a characterization of when $\mathcal{C}$ is privately learnable using *any* finite number of samples whatsoever. The following theorem captures this recent characterization of private learnability.

**Theorem 1** (Informal [Alon et al., 2019, Bun et al., 2020b]). *Let $\mathcal{C}$ be a concept class with* Littlestone *dimension $d = L(\mathcal{C})$. Then $2^{O(d)}$ samples are sufficient to privately learn $\mathcal{C}$ and $\Omega(\log^* d)$ samples are necessary.*

The Littlestone dimension is a combinatorial parameter that exactly captures the complexity of learning $\mathcal{C}$ in Littlestone's mistake bound model of online learning [Littlestone, 1987]. Thus, Theorem 1

characterizes the privately learnable concept classes as exactly those that are online learnable. This is but one manifestation of the close connection between online learning and private algorithm design, e.g., [Roth and Roughgarden, 2010, Dwork et al., 2010, Hardt and Rothblum, 2010, Jain et al., 2012, Feldman and Xiao, 2015, Agarwal and Singh, 2017, Abernethy et al., 2017, Alon et al., 2019, Bousquet et al., 2019, Neel et al., 2019, Gonen et al., 2019, Bun et al., 2020a, Jung et al., 2020] that facilitates the transfer of techniques between the two areas.

Theorem 1 means that, at least in principle, online learning algorithms can be generically converted into differentially private learning algorithms and vice versa. The obstacle, however, is the efficiency of conversion, both in terms of sample use and running time. The forward direction (online learning $\implies$ private learning) [Bun et al., 2020b] gives an algorithm that incurs at least an exponential blowup in both complexities. The reverse direction (private learning $\implies$ online learning) [Alon et al., 2019] is non-constructive: It is proved using the fact that every class with Littlestone dimension at least $d$ contains an embedded copy of $\mathcal{THR}_{\log d}$, the class of one-dimensional threshold functions over a domain of size $\log d$. The characterization then follows from a lower bound of $\Omega(\log^* d)$ on the sample complexity of privately learning $\mathcal{THR}_{\log d}$. (In earlier work, Feldman and Xiao [2015] used a similar connection to show that "pure" $(\varepsilon, 0)$-differentially private learning requires finite Littlestone dimension.)

There is limited room to improve the general sample complexity relationships in Theorem 1. There are classes of Littlestone dimension $d$ (e.g., Boolean conjunctions) that require $\Omega(d)$ samples to learn even non-privately. Meanwhile, $\mathcal{THR}_{2^d}$ has Littlestone dimension $d$ but can be privately learned using $\tilde{O}((\log^* d)^{1.5})$ samples [Kaplan et al., 2020]. This latter result rules out, say, a generic conversion from polynomial-sample private learners to online learners with polynomial mistake bound.

In this work, we address the parallel issue of computational complexity. Could there be a computationally efficient black-box conversion from any private learner to an online learner? We show that, under cryptographic assumptions, the answer is *no*.

**Theorem 2** (Informal, building on [Blum, 1994]). *Assuming the existence of one-way functions, there is a concept class that is privately PAC learnable in polynomial-time, but is not online learnable by any poly-time algorithm with a polynomial mistake bound.*

The question we study was explicitly raised by Neel et al. [2019]. They proved a barrier result to the existence of "oracle-efficient" private learning by showing that a restricted class of such learners can be efficiently converted into oracle-efficient online learners, the latter for which negative results are known [Hazan and Koren, 2016]. This led them to ask whether their barrier result could be extended to all private learning algorithms. Our Theorem 2 thus stands as a "barrier to a barrier" to general oracle-efficient private learning.

This question was also studied in a recent work of Gonen et al. [2019]. They gave an efficient conversion from *pure*-private learners to online learners with sublinear regret. As we discuss in Section 3.6 the efficiency of their construction relies on either a uniform model of learning (that turns out to be incompatible with pure differential privacy) or on an additional assumption that the private learner is highly sample efficient. Theorem 2 rules out the existence of such a conversion even for non-uniform pure-private learners without this extra sample efficiency condition.

**Proof idea of Theorem 2.** In 1990, Blum [1994] defined a concept class we call $\mathcal{OWS}$ that is (non-privately) efficiently PAC learnable but not poly-time online learnable. We prove Theorem 2 by showing that $\mathcal{OWS}$ is efficiently privately PAC learnable as well. Blum's construction builds on the Goldreich-Goldwasser-Micali [Goldreich et al., 1986] pseudorandom function generator to define families of "one-way" sequences $\sigma_1, \ldots, \sigma_r \in \{0, 1\}^d$ for some $r$ that is exponential in $d$. Associated to each $\sigma_i$ is a label $b_i \in \{0, 1\}$. These strings and their labels have the property that they can be efficiently computed in the forward direction, but are hard to compute in the reverse direction. Specifically, given $\sigma_i$ and any $j > i$, it is easy to compute $\sigma_j$ and $b_j$. On the other hand, given $\sigma_j$, it is hard to compute $\sigma_i$ and hard to predict $b_i$.

To see how such sequences can be used to separate online and non-private PAC learning, consider the problem of determining the label $b_i$ for a given string $\sigma_i$. In the online setting, an adversary may present the sequence $\sigma_r, \sigma_{r-1}, \ldots$ in reverse. Then the labels $b_j$ are unpredictable to a poly-time learner. On the other hand, a poly-time PAC learner can identify the $\sigma_{i^*}$ with smallest index in its

sample of size $n$. Then all but a roughly $1/n$ fraction of the underlying distribution on examples will have index at least $i^*$ and can be predicted using the ease of forward computation.

Note that the PAC learner for $\mathcal{OWS}$ is not private, since the classifier based on $i^*$ essentially reveals this sample in the clear. We design a private version of this learner by putting together standard algorithms from the differential privacy literature in a modular way. We first privately identify an $i^*$ that is approximately smallest in the sample. Releasing $\sigma_{i^*}$ directly at this stage is still non-private. So instead, we privately check that every $\sigma_i$ with $i \leq i^*$ corroborates the string $\sigma_{i^*}$, in that $\sigma_{i*}$ is the string that would be obtained by computing forward using any of these strings. If the identity of $\sigma_{i^*}$ is stable in this sense and passes the privacy-preserving check, then it is safe to release.

## 2 Preliminaries

An *example* is an element $x \in \{0, 1\}^d$. A *concept* is a boolean function $c : \{0, 1\}^d \to \{0, 1\}$. A *labeled example* is a pair $(x, c(x))$. A *concept class* $\mathcal{C} = \{\mathcal{C}_d\}_{d \in \mathbb{N}}$ is a sequence where each $\mathcal{C}_d$ is a set of concepts over $\{0, 1\}^d$. Associated to each $\mathcal{C}_d$ is a (often implicit) *representation scheme* under which concepts are encoded as bit strings. Define $|c|$ to be the minimum length representation of $c$.

### 2.1 PAC Learning

In the PAC model, there is an unknown target concept $c \in \mathcal{C}$ and an unknown distribution $\mathcal{D}$ over labeled examples $(x, c(x))$. Given a sample $((x_i, c(x_i)))_{i=1}^n$ consisting of i.i.d. draws from $\mathcal{D}$, the goal of a learning algorithm is to produce a hypothesis $h : \{0, 1\}^d \to \{0, 1\}$ that approximates $c$ with respect to $\mathcal{D}$. Specifically, the goal is to find $h$ with low population loss defined as follows.

**Definition 3** (Population Loss). Let $\mathcal{D}$ be a distribution over $\{0, 1\}^d \times \{0, 1\}$. The population loss of a hypothesis $h : \{0, 1\}^d \to \{0, 1\}$ is

$$\text{loss}_{\mathcal{D}}(h) = \Pr_{(x,b) \sim \mathcal{D}}[h(x) \neq b].$$

Throughout this work, we consider *improper* learning algorithms where the hypothesis $h$ need not be a member of the class $\mathcal{C}$. A learning algorithm $L$ *efficiently PAC learns* $\mathcal{C}$ if for every target concept $c$, every distribution $\mathcal{D}$, and parameters $\alpha, \beta > 0$, with probability at least $1 - \beta$ the learner $L$ identifies a poly-time evaluable hypothesis $h$ with $\text{loss}_{\mathcal{D}}(h) \leq \alpha$ in time $\text{poly}(d, 1/\alpha, 1/\beta, |c|)$. It is implicit in this definition that the number of samples $n$ required by the learner is also polynomial. We will also only consider classes where $|c|$ is polynomial in $d$ for every $c \in \mathcal{C}_d$, so we may regard a class as efficiently PAC learnable if it is learnable using $\text{poly}(d, 1/\alpha, 1/\beta)$ time and samples.

### 2.2 Online Learning

We consider two closely related models of online learning: The mistake-bound model and the no-regret model. Our negative result holds for the weaker no-regret model, making our separation stronger. We first review Littlestone's mistake-bound model of online learning [Littlestone, 1987]. This model is defined via a two-player game between a learner and an adversary. Let $\mathcal{C}$ be a concept class and let $c \in \mathcal{C}$ be chosen by the adversary. Learning proceeds in rounds. In each round $t = 1, \ldots, 2^d$,

    (i) The adversary selects an $x_t \in \{0, 1\}^d$,

    (ii) The learner predicts $\hat{b}_t \in \{0, 1\}$, and

    (iii) The learner receives the correct labeling $b_t = c(x_t)$.

A (deterministic) learning algorithm learns $c$ with mistake bound $M$ if for every adversarial ordering of the examples, the total number of incorrect predictions the learner makes is at most $M$. We say that the learner *efficiently mistake-bound learns* $\mathcal{C}$ if for every $c \in \mathcal{C}$ it has mistake bound $\text{poly}(d, |c|)$ and runs in time $\text{poly}(d, |c|)$ in every round.

We also consider a relaxed model of online learning in which the learner aims to achieve *no-regret*, i.e., err with respect to $c$ in a vanishing fraction of rounds. Let $T$ be a time horizon known to a

randomized learner. The goal of the learner is to minimize its regret, defined by

$$R_T = \sum_{t=1}^{T} \mathbb{I}[\hat{b}_t \neq c(x_t)].$$

We say that a learner *efficiently no-regret learns* $\mathcal{C}$ if there exists $\eta > 0$ such that for every adversary, it achieves $\mathbb{E}[R_T] = \text{poly}(d, |c|) \cdot T^{1-\eta}$ using time $\text{poly}(d, |c|, T)$ in every round. Under this formulation, every efficient mistake-bound learner is also an efficient no-regret learner.

We point out two non-standard features of this definition of efficiency. First, "no-regret" typically requires regret to be sublinear in $T$, i.e., $o(T)$ whereas we require it to be strongly sublinear $T^{1-\eta}$. A stronger condition like this is needed to make the definition nontrivial because a regret upper bound of $T \leq d \cdot T / \log T = \text{poly}(d) \cdot o(T)$ is always achievable in our model by random guessing. Many no-regret algorithms achieve strongly sublinear regret, e.g., the experts/multiplicative weights algorithm and the algorithm of Gonen et al. [2019] that both achieve $\eta = 1/2$. Second, it would be more natural to require the learner to run in time polynomial in $\log T$, the description length of the time horizon, rather than $T$ itself. The relaxed formulation here only makes our lower bounds stronger, and we use it to be consistent with the positive result of Gonen et al. [2019] that runs in time proportional to $T$.

## 2.3 Differential Privacy

**Definition 4** (Differential Privacy). Let $\varepsilon, \delta > 0$. A randomized algorithm $L : X^n \to \mathcal{R}$ is $(\varepsilon, \delta)$-differentially private if for every pair of datasets $S, S'$ differing in at most one entry, and every measurable set $T \subseteq \mathcal{R}$,

$$\Pr[L(S) \in T] \leq e^{\varepsilon} \Pr[L(S') \in T] + \delta.$$

We refer to the special case where $\delta = 0$ as *pure $\varepsilon$-differential privacy*, and the case where $\delta > 0$ as *approximate* differential privacy.

When $L$ is a learning algorithm, we require that this condition hold for all neighboring pairs of samples $S, S'$ – not just those generated according to a distribution on examples labeled by a concept in a given class.

## 3 Theorem 2: Privately PAC Learning One-Way Sequences

### 3.1 One-Way Sequences

For every $d$, Blum defines a concept class $\mathcal{OWS}_d$ consisting of functions over the domain $\{0, 1\}^d$ that can be represented using $\text{poly}(d)$ bits and evaluated in $\text{poly}(d)$ time. The concepts in $\mathcal{OWS}_d$ are indexed by bit strings $s \in \{0, 1\}^k$, where $k = \lfloor \sqrt{d} \rfloor - 1$. The definition of $\mathcal{OWS}_d$ is based on two efficiently representable and computable functions $G : \{0, 1\}^k \times \{0, 1\}^k \to \{0, 1\}^{d-k}$ and $f : \{0, 1\}^k \times \{0, 1\}^k \to \{0, 1\}$ that are based on the Goldreich-Goldwasser-Micali pseudorandom function family [Goldreich et al., 1986]. The exact definition of these functions are not important to our treatment, so we refer the reader to [Blum, 1994] for details. Here, $G(i, s)$ computes the string $\sigma_i$ as described in the introduction, and $f(i, s)$ computes its label $b_i$. For convenience we identify $\{0, 1\}^k$ with $[2^k]$.

We are now ready to define the concept class $\mathcal{OWS}_d = \{c_s\}_{s \in \{0,1\}^k}$ where

$$c_s(i, \sigma) = \begin{cases} 1 & \text{if } \sigma = G(i, s) \text{ and } f(i, s) = 1 \\ 0 & \text{otherwise.} \end{cases}$$

We recall the two key properties of the sequences $\sigma_i$ obtained from these strings. They are easy to compute in the forward direction, even in a random-access fashion, but difficult to compute in reverse. These properties are captured by the following claims.

**Proposition 5** ([Blum, 1994]). *There is an efficiently computable function* ComputeForward : $\{0, 1\}^k \times \{0, 1\}^k \times \{0, 1\}^{d-k} \to \{0, 1\}^{d-k} \times \{0, 1\}$ *such that* ComputeForward$(j, i, G(i, s)) = \langle G(j, s), f(j, s) \rangle$ *for every $j > i$.*

**Proposition 6** ([Blum, 1994], Corollary 3.4). *Suppose $G$ and $f$ are constructed using a secure pseudorandom generator. Let $\mathcal{O}$ be an oracle that, on input $j > i$ and $G(i, s)$ outputs $\langle G(j, s), f(j, s) \rangle$. For every poly-time probabilistic algorithm $A$ and every $i \in \{0, 1\}^k$,*

$$\Pr[A^{\mathcal{O}}(i, G(i, s)) = f(i, s)] \leq \frac{1}{2} + \mathrm{negl}(d),$$

*where the probability is taken over the coins of $A$ and uniformly random $s \in \{0, 1\}^k$.*

Blum used Proposition 6 to show that $\mathcal{OWS}$ cannot be efficiently learned in the mistake bound model (even with membership queries). In the full version of this work, we adapt his argument to the setting of no-regret learning.

**Proposition 7.** *If $G$ and $f$ are constructed using a secure pseudorandom generator, then $\mathcal{OWS}$ cannot be learned by an efficient no-regret algorithm.*

In the rest of this section, we construct an $(\varepsilon, \delta)$-differentially private learner for $\mathcal{OWS}$.

## 3.2   Basic Differential Privacy Tools

The *sensitivity* of a function $q : X^n \to \mathbb{R}$ is the maximum value of $|q(S) - q(S')|$ taken over all pairs of datasets $S, S'$ differing in one entry.

**Lemma 8** (Laplace Mechanism). *The Laplace distribution with scale $\lambda$, denoted $\mathrm{Lap}(\lambda)$, is supported on $\mathbb{R}$ and has probability density function $f_{\mathrm{Lap}(\lambda)}(x) = \exp(-|x|/\lambda)/2\lambda$. If $q : X^n \to \mathbb{R}$ has sensitivity 1, then the algorithm $M_{\mathrm{Lap}}(S) = q(S) + \mathrm{Lap}(1/\varepsilon)$ is $\varepsilon$-differentially private and, for every $\beta > 0$, satisfies $|M_{\mathrm{Lap}}(S) - q(S)| \leq \log(2/\beta)/\varepsilon$ with probability at least $1 - \beta$.*

**Remark 9.** *We describe our algorithm using the Laplace mechanism as a matter of mathematical convenience, even though sampling from the continuous Laplace distribution is incompatible with the standard Turing machine model of computation. To achieve strict polynomial runtimes on finite computers we would use in its place the Bounded Geometric Mechanism [Ghosh et al., 2012, Balcer and Vadhan, 2018].*

**Theorem 10** (Exponential Mechnanism [McSherry and Talwar, 2007]). *Let $q : X^n \times \mathcal{R} \to \mathbb{R}$ be a sensitivity-1 score function. The the algorithm that samples $r \in \mathcal{R}$ with probability $\propto \exp(\varepsilon q(S, r)/2)$ satisfies*

1. *$\varepsilon$-differential privacy, and*

2. *For every $S$, with probability at least $1 - \beta$ the sampled $\hat{r}$ satisfies*

$$q(S, \hat{r}) \geq \max_{r \in \mathcal{R}} q(S, r) - \frac{2 \log(|\mathcal{R}|/\beta)}{\varepsilon}.$$

The following "basic" composition theorem allows us to bound the privacy guarantee of a sequence of adaptively chosen algorithms run over the same dataset.

**Lemma 11** (Composition, e.g., [Dwork and Lei, 2009]). *Let $M_1 : X^n \to \mathcal{R}_1$ be $(\varepsilon, \delta)$-differentially private. Let $M_2 : X^n \times \mathcal{R}_1 \to \mathcal{R}_2$ be $(\varepsilon_2, \delta_2)$ differentially private for every fixed value of its second argument. Then the composition $M(S) = M_2(S, M_1(S))$ is $(\varepsilon_1 + \varepsilon_2, \delta_1 + \delta_2)$-differentially private.*

## 3.3   Private Robust Minimum

**Definition 12.** Given a dataset $S = (x_1, \ldots, x_n) \in [R]^n$, an $\alpha$-*robust minimum* for $S$ is a number $r \in [R]$ such that

1. *$|\{i : x_i \leq r\}| \geq \alpha n$, and*

2. *$|\{i : x_i \leq r\}| \leq 2\alpha n$.*

Note that $r$ need not be an element of $S$ itself – this is important for ensuring that we can release a robust minimum privately. Condition 2 guarantees that $r$ is approximately the minimum of $S$. Condition 1 guarantees that this condition holds robustly, i.e., one needs to change at least $\alpha n$ points of $S$ before $r$ fails to be at least the minimum.

**Theorem 13.** *There exist polynomial-time algorithms* Min$_{\text{pure}}$ *and* Min$_{\text{approx}}$ *that each solve the private robust minimum problem with probability at least* $1 - \beta$, *where*

1. *Algorithm* Min$_{\text{pure}}$ *is* $\varepsilon$-*differentially private and succeeds as long as*

$$n \geq O\left(\frac{\log(R/\beta)}{\alpha\varepsilon}\right).$$

2. *Algorithm* Min$_{\text{approx}}$ *is* $(\varepsilon, \delta)$-*differentially private and succeeds as long as*

$$n \geq \tilde{O}\left(\frac{(\log^* R)^{1.5} \cdot \log^{1.5}(1/\delta) \cdot \log(1/\beta)}{\alpha\varepsilon}\right).$$

*Proof.* The algorithms are obtained by a reduction to the *interior point problem.* Both this problem and essentially the same reduction are described in [Bun et al., 2015] but we give the details for completeness. In the interior point problem, we are given a dataset $S \in [R]^n$ and the goal is to identify $r \in [R]$ such that $\min S \leq r \leq \max S$. An $(\varepsilon, \delta)$-DP algorithm that solves the interior point problem using $m$ samples and success probability $1 - \beta$ can be used to solve the robust minimum problem using $n = O(m/\alpha)$ samples: Given an instance $S$ of the robust minimum problem, let $S'$ consist of the elements $\lceil \alpha n \rceil$ through $\lfloor 2\alpha n \rfloor$ of $S$ in sorted order and apply the interior point algorithm to $S'$.

The exponential mechanism provides a pure $\varepsilon$-DP algorithm for the interior point problem with sample complexity $O(\log(R/\beta)/\varepsilon)$ [Smith, 2011]. Let $x_{(1)}, \ldots, x_{(n)}$ denote the elements of $S$ in sorted order. The appropriate score function $q(S, r)$ is the maximum value of $\min\{t, n - t\}$ such that $x_{(1)} \leq \cdots \leq x_{(t)} \leq r \leq x_{(t+1)} \leq \cdots \leq x_{(n)}$. Thus $q(S, r)$ ranges from a maximum of $\lfloor n/2 \rfloor$ iff $r$ is a median of $S$ to a minimum of $0$ iff $r$ is not an interior point of $S$. By Theorem 10, the released point has positive score (and hence is an interior point) as long as $n > 4\log(R/\beta)/\varepsilon$. Moreover, one can efficiently sample from the exponential mechanism distribution in this case as the distribution is constant on every interval of the form $[x_{(t)}, x_{(t+1)})$.

For $(\varepsilon, \delta)$-DP with $\delta > 0$, Kaplan et al. [2020] provide an efficient algorithm for the interior point problem (with constant failure probability) using $\tilde{O}((\log^* R)^{1.5} \log^{1.5}(1/\delta)/\varepsilon)$ samples. Taking the median of $O(\log(1/\beta))$ repetitions of their algorithm on disjoint random subsamples gives the stated bound. $\qquad\square$

## 3.4 Private Most Frequent Item

Let $S \in X^n$ be a dataset and let $x \in X$ be the item appearing most frequently in $S$. The goal of the "private most-frequent-item problem" is to identify $x$ with high probability under the assumption that the most frequent item is stable: its identity does not change in a neighborhood of the given dataset $S$. For $x \in X$ and $S = (x_1, \ldots, x_n) \in X^n$, define $\text{freq}_S(x) = |\{i : x_i = x\}|$.

**Definition 14.** An algorithm $M : X^n \to [R]$ solves the most-frequent-item problem with gap GAP and failure probability $\beta$ if the following holds. Let $S \in X^n$ be any dataset with $x^* = \text{argmax}_x \text{freq}_S(x)$ and

$$\text{freq}_S(x^*) \geq \max_{x \neq x^*} \text{freq}_S(x) + \text{GAP}.$$

Then with probability at least $1 - \beta$, we have $M(S) = x^*$.

**Theorem 15** ([Balcer and Vadhan, 2018]). *There exist polynomial-time algorithms* Freq$_{\text{pure}}$ *and* Freq$_{\text{approx}}$ *that each solve the private most-frequent-item problem with probability at least* $1 - \beta$, *where*

1. *Algorithm* Freq$_{\text{pure}}$ *is* $\varepsilon$-*differentially private and succeeds as long as* GAP $\geq O(\log(R/\beta)/\varepsilon)$.

2. *Algorithm* Freq$_{\text{approx}}$ *is* $(\varepsilon, \delta)$-*differentially private and succeeds as long as* GAP $\geq O(\log(n/\delta\beta)/\varepsilon)$.

Balcer and Vadhan [2018] actually solved the more general problem of computationally efficient private histogram estimation. Theorem 15 follows from their algorithm by reporting the privatized bin with the largest noisy count.

## 3.5 Privately Learning $\mathcal{OWS}_d$

---

**Algorithm 1** Pure Private Learner for $\mathcal{OWS}_d$

---

1. Let $S_+ = ((i_1, \sigma_1), \ldots, (i_m, \sigma_m))$ be the subsequence of positive examples in $S$, where $i_1 \leq i_2 \leq \ldots \leq i_m$.

2. Let $\hat{m} = m + \mathrm{Lap}(3/\varepsilon)$. If $\hat{m} \leq \alpha n/3$, output the all-0 hypothesis.

3. Let $I = (i_1, \ldots, i_m)$. Run $\mathsf{Min_{pure}}(I)$ using privacy parameter $\varepsilon/3$ to identify a $(\alpha n/6\hat{m})$-robust minimum $i^*$ of $I$ with failure probability $\beta/6$.

4. For every $i_j \in I$ with $i_j < i^*$ let $\langle \hat{\sigma}_j, \hat{b}_j \rangle = \mathsf{ComputeForward}(i^*, i_j, \sigma_j)$. For every $i_j \in I$ with $i_j = i^*$ let $\langle \hat{\sigma}_j, \hat{b}_j \rangle = \langle \sigma_j, b_j \rangle$.

5. Run $\mathsf{Freq_{pure}}(\langle \hat{\sigma}_1, \hat{b}_1 \rangle, \ldots, \langle \hat{\sigma}_\ell, \hat{b}_\ell \rangle)$ using privacy parameter $\varepsilon/3$ to output $\langle \sigma^*, b^* \rangle$ with failure probability $\beta/6$. Here, $\ell$ is the largest $j$ for which $i_j \leq i^*$.

6. Return the hypothesis $h(i, \sigma) = $
   "If $i < i^*$, output 0. If $i = i^*$, output $b^*$ if $\sigma^* = \sigma$ and output 0 otherwise. If $i > i^*$, run algorithm $\mathsf{ComputeForward}(i, i^*, \sigma^*) = \langle \hat{\sigma}, \hat{b} \rangle$. If $\sigma = \hat{\sigma}$, output $\hat{b}$. Else, output 0."

---

**Theorem 16.** *Algorithm 1 is an $\varepsilon$-differentially private and $(\alpha, \beta)$-PAC learner for $\mathcal{OWS}_d$ running in time* $\mathrm{poly}(d, 1/\alpha, \log(1/\beta))$ *using*

$$n = O\left(\frac{\sqrt{d} + \log(1/\beta)}{\alpha\varepsilon}\right)$$

*samples.*

*Proof.* Algorithm 1 is an adaptive composition of three $(\varepsilon/3)$-differentially private algorithms, hence $\varepsilon$-differentially private by Lemma 11.

To show that it is a PAC learner, we first argue that the hypothesis produced achieves low error with respect to the sample $S$, and then argue that it generalizes to the underlying distribution. That is, we first show that for every realizable sample $S$, with probability at least $1 - \beta/2$ over the randomness of the learner alone, the hypothesis $h$ satisfies

$$\mathrm{loss}_S(h) = \frac{1}{n} \sum_{k=1}^{n} \mathbb{I}[h(x_k) \neq b_k] \leq \alpha/2.$$

We consider several cases based on the number of positive examples $m = |S_+|$. First suppose $m \leq \alpha n/4$. Then Lemma 8 guarantees that the algorithm outputs the all-0 hypothesis with probability at least $1 - \beta$ in Step 2, and this hypothesis has sample loss at most $\alpha/4$.

Now suppose $m = |S_+| \geq \alpha n/2$. Then with probability at least $1 - \beta/3$ we have $|\hat{m} - m| \leq 3\log(6/\beta)/\varepsilon$. In particular this means $\hat{m} \geq \alpha n/3$, so the algorithm continues past Step 2. Now with probability at least $1 - \beta/3$, Step 3 identifies a point $i^*$ such that $|\{i \in I : i \leq i^*\}| \geq (\alpha n/6\hat{m}) \cdot m \geq \alpha n/10$ and $|\{i \in I : i \leq i^*\}| \leq (\alpha n/6\hat{m}) \cdot m \leq \alpha n/2$, as long as

$$m \geq O\left(\frac{k + \log(1/\beta)}{\alpha\varepsilon} \cdot \frac{\hat{m}}{n}\right) \iff n \geq O\left(\frac{\sqrt{d} + \log(1/\beta)}{\alpha\varepsilon}\right).$$

The first condition, in particular, guarantees that $\ell \geq \alpha n/10$. Realizability of the sample $S$ guarantees that the points $\langle \hat{\sigma}_1, \hat{b}_1 \rangle, \ldots, \langle \hat{\sigma}_\ell, \hat{b}_\ell \rangle$ are all identical. So with the parameter $\mathrm{GAP} = \ell \geq \alpha n/10$, Step 5 succeeds in outputting their common value $\langle \sigma^*, b^* \rangle$ with probability at least $1 - \beta/3$, again as long as $n \geq O((\sqrt{d} + \log(1/\beta))/\alpha\varepsilon)$.

We now argue that the hypothesis $h$ produced in Step 6 succeeds on all but $\alpha n/2$ examples. A case analysis shows that the only input samples on which $h$ makes an error are those $(i_j, \sigma_j) \in S_+$ for which $i_j < i^*$. The success criterion of Step 3 ensures that the number of such points is at most $\alpha n/2$.

The final case where $\alpha n/4 < m < \alpha n/2$ is handled similarly, except it is now also acceptable for the algorithm to terminate early in Step 2, outputting the all-0 hypothesis.

We now argue that achieving low error with respect to the sample is sufficient to achieve low error with respect to the distribution: If the learner above achieves sample loss $\mathrm{loss}_S(h) \leq \alpha/2$ with probability at least $1 - \beta/2$, then it is also an $(\alpha, \beta)$-PAC learner for $\mathcal{OWS}_d$ when given at least $n \geq 8\log(2/\beta)/\alpha$ samples. The analysis follows the standard generalization argument for one-dimensional threshold functions, and our presentation follows Bun et al. [2015].

Fix a realizable distribution $\mathcal{D}$ (labeled by concept $c_s$) and let $\mathcal{H}$ be the set of hypotheses that the learner could output given a sample from $\mathcal{D}$. That is, $\mathcal{H}$ consists of the all-0 hypothesis and every hypothesis of the form $h_{i^*}$ as constructed as in Step 6. We may express the all-0 hypothesis as $h_{2^d+1}$. It suffices to show that for a sample $S$ drawn i.i.d. from a realizable distribution $\mathcal{D}$ that

$$\Pr[\exists h \in \mathcal{H} : \mathrm{loss}_\mathcal{D}(h) \leq \alpha \text{ and } \mathrm{loss}_S(h) \leq \alpha/2] \leq \beta/2.$$

Let $i_-$ be the largest number such that $\mathrm{loss}_\mathcal{D}(h_{i_-}) > \alpha$. If some $h_i$ has $\mathrm{loss}_\mathcal{D}(h_i) > \alpha$ then $i \leq i_-$, and hence for any sample, $\mathrm{loss}_S(h_{i-}) \leq \mathrm{loss}_S(h_i)$. So it suffices to show that

$$\Pr[\mathrm{loss}_S(h_{i-}) \leq \alpha/2] \leq \beta/2.$$

Define $E = \{(i, G(i, s)) : i < i_- \text{ and } f(i, s) = 1\}$ to be the set of examples on which $h_{i_-}$ makes a mistake. By a Chernoff bound, the probability that after $n$ independent samples from $\mathcal{D}$, fewer than $\alpha n/2$ appear in $E$ is at most $\exp(-\alpha n/8) \leq \beta/2$ provided $n \geq 8\log(2/\beta)/\alpha$.

$\square$

The same argument, replacing the use of $\mathsf{Min}_{\mathsf{pure}}$ with $\mathsf{Min}_{\mathsf{approx}}$ and $\mathsf{Freq}_{\mathsf{pure}}$ with $\mathsf{Freq}_{\mathsf{approx}}$ in Algorithm 1 yields

**Theorem 17.** *There is an $(\varepsilon, \delta)$-differentially private and $(\alpha, \beta)$-PAC learner for $\mathcal{OWS}_d$ running in time $\mathrm{poly}(d, 1/\alpha, \log(1/\beta))$ using*

$$n = \tilde{O}\left(\frac{(\log^* d)^{1.5} \cdot \log^{1.5}(1/\delta) \cdot \log(1/\beta)}{\alpha\varepsilon}\right)$$

*samples.*

### 3.6 Comparison to Work of Gonen et al. [2019]

Gonen et al. [2019] proved a positive result giving conditions under which pure-private learners can be efficiently compiled into online learners. The purpose of this section is to describe their model and result and, in particular, explain why it does not contradict Theorem 16.

**The [Gonen et al., 2019] reduction and uniform learning.** The reduction of Gonen et al. [2019] works as follows. Let $\mathcal{C}$ be a concept class that is pure-privately learnable (with fixed constant privacy and accuracy parameters) using $m_0$ samples. Consider running this algorithm roughly $N = \exp(m_0)$ times on a fixed dummy input, producing hypotheses $h_1, \ldots, h_N$. Pure differential privacy guarantees that for every realizable distribution on labeled examples, with high probability one of these hypotheses $h_i$ will have small loss. This idea can be used to construct an online learner for $\mathcal{C}$ by treating the random hypotheses $h_1, \ldots, h_N$ as experts and running multiplicative weights to achieve no-regret with respect to the best one online.

As it is described by Gonen et al. [2019], this is a computationally efficient reduction from no-regret learning to *uniform* pure-private PAC learning. In the uniform PAC model, there is a single infinite concept class $\mathcal{C}$ consisting of functions $c : \{0, 1\}^* \to \{0, 1\}$. An efficient uniform PAC learner for $\mathcal{C}$ uses $m(\alpha, \beta)$ samples to learn a hypothesis with loss at most $\alpha$ and failure probability $\beta$ in time $\mathrm{poly}(|c|, 1/\alpha, 1/\beta)$. Note that the number of samples $m(\alpha, \beta)$ is completely independent of the target concept $c$. This contrasts with the non-uniform model, where the number of samples is allowed to grow with $d$, the domain size of $c$.

Another noteworthy difference comes when we introduce differential privacy. In the uniform model, one can move to a neighboring dataset by changing a single entry to any element of $\{0, 1\}^*$. In the non-uniform model, on the other hand, an entry may only change to another element of the same

$\{0, 1\}^d$. This distinction affects results for pure-private learning, as we will see below. However, it does not affect $(\varepsilon, \delta)$-DP learning, as one can always first run the algorithm described in Theorem 15 to privately check that all or most of the elements in the sample come from the same $\{0, 1\}^d$.

A simple example to keep in mind when considering feasibility of learning in the uniform model is the class of point functions $\mathcal{POINT} = \{p_x : x \in \{0, 1\}^*\}$ where $p_x(y) = 1$ iff $x = y$. This class is efficiently uniformly PAC-learnable using $m(\alpha, \beta) = O(\log(1/\beta)/\alpha)$ samples by returning $p_x$ where $x$ is any positive example in the dataset.

**Impossibility of efficinet uniform pure-private learning.** The class $\mathcal{POINT}$ turns out to be uniformly PAC-learnable with pure differential privacy as well [Beimel et al., 2019]. However, this algorithm is *not* computationally efficient. The following claim shows that this is inherent, as indeed it is even impossible to uniformly learn $\mathcal{POINT}$ using hypotheses with short description lengths.

**Proposition 18.** *Let $L$ be a pure $1$-differentially private algorithm for uniformly $(1/2, 1/2)$-PAC learning $\mathcal{POINT}$. Then for every labeled sample $S$, we have $\mathbb{E}_{h \leftarrow L(S)}[|h|] = \infty$.*

In fact, in the full version of this work, we show that *every* infinite concept class is impossible to learn uniformly with pure differential privacy:

**Proposition 19.** *Let $L$ be a pure $1$-differentially private algorithm for uniformly $(1/2, 1/2)$-PAC learning an infinite concept class $\mathcal{C}$. Then for every labeled sample $S$, we have $\mathbb{E}_{h \leftarrow L(S)}[|h|] = \infty$.*

**Sample-efficient learning.** At first glance, this may seem to make the construction of Gonen et al. [2019] vacuous. However, it is still of interest as it can be made to work in non-uniform model of pure-private PAC learning under the additional assumption that the pure-private learner is highly sample efficient. That is, if $\mathcal{C}_d$ is learnable using $m(d) = O(\log d)$ samples, then the number of experts $N$ remains polynomial. There *is* indeed a computationally efficient non-uniform pure private learner for $\mathcal{POINT}$ with sample complexity $O(1)$ [Beimel et al., 2019] that can be transformed into an efficient online learner using their algorithm. This does not contradict our negative result Theorem 16, as that pure-private learner uses sample complexity $O(\sqrt{d})$.

# 4 Conclusion

In this paper, we showed that under cryptographic assumptions, efficient private learnability does not necessarily imply efficient online learnability. Our work raises a number of additional questions about the relationship between efficient learnability between the two models.

**Uniform approximate-private learning.** In Section 3.6 we discussed the uniform model of (private) PAC learning and argued that efficient learnability is impossible under pure privacy. It is, however, possible under approximate differential privacy, e.g., for point functions. Thus it is of interest to determine whether uniform approximate-private learners can be efficiently transformed into online learners. Our negative result for non-uniform learning uses sample complexity $\tilde{O}((\log^* d)^{1.5})$ to approximate-privately learn the class $\mathcal{OWS}_d$, so it does not rule out this possibility.

**Efficient conversion from online to private learning.** Is a computationally efficient version of [Bun et al., 2020b] possible? Note that to exhibit a concept class $\mathcal{C}$ refuting this, $\mathcal{C}$ must in particular be efficiently PAC learnable but not efficiently privately PAC learnable. There is an example of such a class $\mathcal{C}$ based on "order-revealing encryption" [Bun and Zhandry, 2016]. However, a similar adversary argument as what is used for $\mathcal{OWS}$ can be used to show that this class $\mathcal{C}$ is also not efficiently online learnable.

**Agnostic private vs. online learning.** Finally, we reiterate a question of Gonen et al. [2019] who asked whether efficient *agnostic* private PAC learners can be converted to efficient agnostic online learners.

## Broader Impact

As this work is theoretical in nature, its tangible ethical and societal impacts are especially difficult to predict. Optimistically, the conceptual message that efficient private learning is possible without efficient online learning opens the door to the design of private learners for classes for which there are no efficient online counterparts. This could lead to surprising tractable learning algorithms for problems motivated by the practice of DP, and downstream, enable the analysis of data that could otherwise not be shared.

Computational efficiency in differential privacy is a major bottleneck for its real-world adoption, and must often be traded off against statistical error and bias. The availability of new, more efficient tools should ideally only increase the adoption and efficacy of privcy-preserving technology. However, there is also a general danger of making inappropriate choices of these tradeoffs when new technologies become available. For instance, a much more efficient algorithm with weaker privacy or accuracy guarantees might be chosen over a less efficient one for, e.g., a real-time analytics application. The availability of new tools should always be complemented by thorough analyses of their accuracy and privacy guarantees so that such decisions can be made in an informed and careful manner.

## Acknowledgments and Disclosure of Funding

I thank Shay Moran for helpful discussions concerning the relationship of this work to [Gonen et al., 2019].

This work was supported by NSF grant CCF-1947889. Within the 36 months prior to the submission of this work, the author was supported by a Google Research Fellowship at the Simons Institute for the Theory of Computing.

## Footnotes

*The full version of this work appears at https://arxiv.org/abs/2007.05665.

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
