[Reviews · NeurIPS 2020]

Review 1

Summary and Contributions: This work shows that there is a class that is privately PAC learnable in polynomial time, but not efficiently learnable (assuming the existence of one-way functions) in the online setting, i.e., there is no polynomial time algorithm with a polynomial mistake bound. A line of recent work (focused on sample complexity) has demonstrated various equivalences between private PAC and online learning; this result focuses on the question of efficiency, proving that efficient private learnability does not imply efficient online learnability if one-way functions exist. The cryptographic assumption is standard for such results and is needed when ruling out general polynomial time learners. The class of functions that cannot be efficiently learned in the online model was given by [Blum 1994], which showed a separation between distribution free PAC learning and online learning. Much of the work toward separating the models of learning, which involves working out the cryptographic construction and giving an efficient PAC algorithm, was done there -- the technical contribution here is to give a private version of that algorithm. That algorithm is not private in that the resulting hypothesis reveals a ‘minimum’ value of the dataset and a key example; privacy is achieved here by the careful composition of existing algorithms to privately select a minimum and key example. Ensuring privacy incurs a dependence on the domain size in the sample complexity.

Strengths: This work settles (using a minimal and widely accepted assumption) the question of an efficient reduction from online learning to private PAC learning in the negative. Though just recently considered, as sample equivalences are only recently known, it is a natural and fundamental question studied by a few works (Gonen et al 2019 and Neel et al 2019). The proof is relatively simple (in a good way) and builds nicely on the work of Blum. It also complements and clarifies the result of Gonen et al. 2019. The discussion around the uniformity/efficient sample complexity requirement for their reduction could be important/useful for other researchers.

Weaknesses: There is not a lot of technical novelty here, as the result is obtained by combining Blum’s work with existing privacy algorithms. I find this to be a minor weakness, since the result answers a fundamental question and is well presented and executed. Update after author response: ======================= After the author's response my overall view of the paper , which is positive, is the same, and I'm glad they woul include the details of the proof in the final version.

Correctness: The claims in the paper are correct; some details of the proofs are deferred to a full version and I hope the authors will provide an appendix with the full argument in the next round (that being said the omitted parts are quite believable).

Clarity: The paper is generally clear enough. A few points of confusion I had: - I’m not clear on the indexing of sigma in the description of Algorithm 1. Shouldn’t the non-hat sigmas be indexed by their corresponding i, the place in the OWS? At least this seems to be the convention at the end of step 4. In that case sigma_1 should be sigma_{i_1} etc in step 1. - Line 228 should it be x*? - It might be helpful to explicitly point out that an example (i,sigma) has i in {0,1}^k when describing the OWS, and then to make some statement in proof of Thm 16 about the number of samples, i.e., since Min_pure is run over a domain of size 2^k=2^{sqrt{d}} and Freq_pure, etc. It sounds from the proof like only (8/alpha) log 2/beta samples are needed.

Relation to Prior Work: The paper’s relationship to previous work is well and thoroughly explained. One question I had is whether the work of Feldman and Xiao 2015 isn’t also relevant around lines 52-58 (don’t they show that pure DP learning sample complexity d implies Littlestone dimension O(d))?

Reproducibility: Yes

Additional Feedback:


Review 2

Summary and Contributions: The paper proves that, assuming the existence of one-way function, there exists a class of hypotheses that is poly-time private PAC learnable but is not poly-time online learnable. The proof is via an easy adaptation of an analogous separation of (non-private) PAC and online.

Strengths: Simple proof + resolves an open problem from FOCS paper!

Weaknesses: The result, which the author describes as “barrier to barrier” is very theoretical and I’m not sure if NeurIPS is the best venue?

Correctness: Yes

Clarity: Yes

Relation to Prior Work: Yes. *** OBSOLETE *** There is an existing huge separation in terms of sample complexity, albeit for non-efficient algorithms. I’m wondering whether that example can be padded to make the algorithms “polynomial” time? [Thanks author for clarifying this point! If I understand correctly, the issue is that an unconditional exponential separation is formally impossible since Littlestone's dimension is bounded by log(|H|) \le log(|U|^VC) = [sample description length] * [complexity of PAC learning]. Right?]

Reproducibility: Yes

Additional Feedback: Typos: Line 30: “C is *privately* learnable” Line 67: “the latter for which”


Review 3

Summary and Contributions: This paper studies the relationship of private PAC learning and online learning, from a computational complexity aspect. It asks whether every class that is privately PAC learnable in polynomial time is also online learnable in polynomial time. The authors give a negative answer to this question, by providing a class that is privately learnable in poly time (giving both pure and approximate learners) but not online learnable, under the existence of one-way functions. They also explore the connection of this result with Gonen et al. that appeared in NeurIPS 2019, which gave a computationally efficient reduction from online learning to uniform pure-private learning (a strong private learning setting over an infinite concept class, with sample complexity independent of the size of the concept). In this paper, the authors prove that such uniform pure-private learners could not be computationally efficient. However, they point to a relaxation of uniform pure-private learners to highly sample-efficient pure-private learners, that would still allow the application of the reduction of Gonen et al.

Strengths: -Recent results on the equivalence of online and private learning left important open questions on whether the sample and computational complexity gaps that these general reductions incur can be reduced. The specific question of whether computationally efficient private learning implies efficient online learning, was also explicitly stated as an interesting open problem in a FOCS 2019 paper. This paper negatively resolves it. -The complimentary results in relation to the Gonen et al. paper, further clarify the space to direct future research to classes or private learning models where these reductions could be efficient.

Weaknesses: -There is a slight drawback, which is that the authors have omitted to submit the supplementary material. However, I find the details provided in the main body to be enough to establish the correctness of the results with sufficient confidence (in particular, the private learning algorithms are fully analyzed, and the impossibility of online learning this class is largely based on Blum1994. As this is the main contribution of the paper, I don't consider this to be significant.

Correctness: All claims are either fully proven or sufficient evidence is given to be convinced of their correctness.

Clarity: The paper is well written.

Relation to Prior Work: Yes.

Reproducibility: Yes

Additional Feedback: The paper was overall very well written. I think that having a section (section 4) to explore the complicated relationship between this paper and Gonen et al was a good idea as it gives a thorough exploration of how these results could be interpreted. But I do think it is a bit harder to read. Perhaps breaking it into three parts (Gonen et al and what is uniform pure private learning, impossibility of efficient uniform pure-private learning, and relaxations that allow the reduction to be useful) would help. Minor typos: -Line 215: is q(S,r) is -> , q(S,r), is -
Line 223: identify -> identity ========================== Thank you for your response. I am keeping my score as is and support acceptance of this paper.

[Author Response · NeurIPS 2020]

Dear reviewers and area chairs,

Thank you for the careful reading of our manuscript and for the helpful comments and suggestions. We will address all
of the smaller suggestions for improving the next revision of the manuscript. Below, we respond to some of the more
significant points the reviewers raised.

**Reviewer 1.** *… some details of the proofs are deferred to a full version …*

We apologize for not correctly uploading the full version of the paper as supplementary material as we had intended.
The full version will be included with the final submission. At the bottom of this response, please find a proof of the
unsubstantiated Proposition 18 (excerpted from the public full version).

*One question I had is whether the work of Feldman and Xiao 2015 isn't also relevant around lines 52-58 …*

Feldman and Xiao's result is indeed highly relevant to our discussion about private sample complexity vs. mistake
bound. We will explain this in the next revision.

**Reviewer 2.** *The result, which the author describes as "barrier to barrier" is very theoretical and I'm not sure if*
*NeurIPS is the best venue?*

The "barrier to a barrier" interpretation of our result places it in the context of an explicit question raised by prior work.
Beyond this context, however, our result addresses one of the most basic questions about learning in two fundamental
and well-studied models. We believe that this puts it in scope for a broad and inclusive NeurIPS community.

*There is an existing huge separation in terms of sample complexity, albeit for non-efficient algorithms. I'm wondering*
*whether that example can be padded to make the algorithms "polynomial" time?*

The challenge is that known sample complexity separations hold for classes that *do* have efficient algorithms. One-
dimensional thresholds over a domain of size $d$ can be efficiently, privately PAC learned using $\approx \log^* d$ samples and
efficiently online learned using $\log d$ samples (via binary search). Blum's class does as the reviewer suggests, using
cryptography to amplify the hardness of online learning thresholds while keeping PAC learning easy. Since private
learning implies non-private learning, any way to obtain our result would imply Blum's result that efficient non-private
learning $\not\Rightarrow$ efficient online learning; to our knowledge, Blum's class is the simplest one that achieves this.

**Reviewer 3.** *There is a slight drawback, which is that the authors have omitted to submit the supplementary material.*

Please see our response to Reviewer 1 and the material at the bottom of this page.

*Perhaps breaking it into three parts (Gonen et al and what is uniform pure private learning, impossibility of efficient*
*uniform pure-private learning, and relaxations that allow the reduction to be useful) would help.*

This is a great suggestion for improving the readability of this section and will be incorporated in the next revision.

*Proof of Proposition 18.* Let $t > 0$. We will show that $\mathbb{E}[|h|] \geq t$. Let $n$ be the number of samples used by $L$. Let $\mathcal{H}_t$
be the set of all functions $h : \{0,1\}^* \to \{0,1\}$ with description length $|h| \leq 2e^n t$. Lemma 1 below shows that there
exists a concept $c \in \mathcal{C}$ and a pair $x, y$ such that $c(x) = 1$ and $c(y) = 0$ but $h(x) = 0$ or $h(y) = 1$ for every $h \in \mathcal{H}_t$.

Consider the distribution $\mathcal{D}$ that is uniform over $(x, 1)$ and $(y, 0)$. Accuracy of the learner requires that $\Pr_{S' \sim \mathcal{D}^n}[L(S') \notin$
$\mathcal{H}_t] \geq 1/2$. Since any sample $S'$ can be obtained from $S$ by changing at most $n$ elements of $S$, pure differential privacy
implies that $\Pr[L(S) \notin \mathcal{H}_t] \geq e^{-n}/2$. Hence $\mathbb{E}_{h \leftarrow L(S)}[|h|] \geq 2e^n t \cdot e^{-n}/2 \geq t$ as we wanted to show. $\square$

Let $\mathcal{S} = \{S_1, \ldots, S_n\}$ be a collection of subsets of $\{0,1\}^*$. We say that $\mathcal{S}$ *generates* another set $T \subseteq \{0,1\}^*$ if for
every pair $x, y \in \{0,1\}^*$ with $x \in T$ and $y \notin T$, there exists $i \in [n]$ such that $x \in S_i$ and $y \notin S_i$.

**Lemma 1.** *A collection $\mathcal{S} = \{S_1, \ldots, S_n\}$ generates at most $2^{2^n}$ distinct sets $T \subseteq \{0,1\}^*$.*

*Proof.* By doubling the size of $\mathcal{S}$ we may assume it is closed under complement, i.e., $S \in \mathcal{S}$ iff $\overline{S} \in \mathcal{S}$. Let us say that a
set $R \subseteq \{0,1\}^*$ is *pairwise separated* by $\mathcal{S}$ if for every pair $x, y \in R$, there exists $i \in [n]$ such that $x \in S_i$ and $y \notin S_i$.
Let $r$ denote the maximum size of a set that is pairwise separated by $\mathcal{S}$; by induction, $r \leq 2^{n-1}$. We will show that if
$T$ is generated by $\mathcal{S}$, then determining the membership of each element of $R$ in $T$ completely determines the set $T$.
Therefore, there are at most $2^r \leq 2^{2^{n-1}}$ possible choices for $T$.

To see this, suppose for the sake of contradiction that there are two sets $T_1, T_2$ that are generated by $\mathcal{S}$ for which
$T_1 \cap R = T_2 \cap R := I$. Let $z$ be an element on which $T_1, T_2$ disagree; say $z \in T_1$ but $z \notin T_2$. We derive our
contradiction by showing that $R \cup \{z\}$ is pairwise separated by $\mathcal{S}$, contradicting the maximality of $R$. To do so, all we
need to show is that for every $y \in R$, there exists $S_i$ such that $z \in S_i$ and $y \notin S_i$, and that there exists $S_j$ such that
$z \notin S_j$ and $y \in S_j$. If $y \in I$, we can take $S_i$ to be the set such that $z \notin \overline{S_i}$ and $y \in \overline{S_i}$ as guaranteed by the fact that
$\mathcal{S}$ generates $T_2$. If $y \notin I$, we can take $S_i$ to be the set such that $z \in S_i$ and $y \notin S_i$ as guaranteed by the fact that $\mathcal{S}$
generates $T_1$. A similar argument can be used to construct $S_j$. $\square$

[Meta-Review · NeurIPS 2020]

This work studies the relationship between private PAC learning and online learning from a computational perspective. The paper establishes a computational separation between the two problems by showing a class that is efficiently privately PAC learnable but not efficiently learnable in the online setting. This is an important theoretical result that resolves an open question in prior work, and will be of interest to the community.